# Efficient Adversarial Training without Attacking: Worst-Case-Aware Robust Reinforcement Learning

**Yongyuan Liang**[†][*] **Yanchao Sun**[‡][*] **Ruijie Zheng**[‡] **Furong Huang**[‡]
[†] Shanghai AI Lab, [‡] University of Maryland, College Park
[†]cheryllLiang@outlook.com [‡]{ycs,rzheng12,furongh}@umd.edu

## Abstract

Recent studies reveal that a well-trained deep reinforcement learning (RL) policy can be particularly vulnerable to adversarial perturbations on input observations. Therefore, it is crucial to train RL agents that are robust against any attacks with a bounded budget. Existing robust training methods in deep RL either treat correlated steps separately, ignoring the robustness of long-term rewards, or train the agents and RL-based attacker together, doubling the computational burden and sample complexity of the training process. In this work, we propose a strong and efficient robust training framework for RL, named Worst-case-aware Robust RL (WocaR-RL), that directly estimates and optimizes the worst-case reward of a policy under bounded $\ell_p$ attacks without requiring extra samples for learning an attacker. Experiments on multiple environments show that WocaR-RL achieves state-of-the-art performance under various strong attacks, and obtains significantly higher training efficiency than prior state-of-the-art robust training methods. The code of this work is available at https://github.com/umd-huang-lab/WocaR-RL.

## 1 Introduction

Deep reinforcement learning (DRL) has achieved impressive results by using deep neural networks (DNN) to learn complex policies in large-scale tasks. However, well-trained DNNs may drastically fail under adversarial perturbations of the input [1, 6]. Therefore, before deploying DRL policies to real-life applications, it is crucial to improve the robustness of deep policies against adversarial attacks, especially worst-case attacks that maximally depraves the performance of trained agents [42].

A line of regularization-based robust methods [54, 33, 40] focuses on improving the robustness of the DNN itself and regularizes the policy network to output similar actions under bounded state perturbations. However, different from supervised learning problems, the vulnerability of a deep policy comes not only from the DNN approximator, but also from the dynamics of the RL environment [52]. These regularization-based methods neglect the intrinsic vulnerability of policies under the environment dynamics, and thus may still fail under strong attacks [42]. For example, in the go-home task shown in Figure 1, both the green policy and the red policy arrive home without rock collision, when there is no attack. However, although regularization-based methods may ensure a minor action change under a state perturbation, the red policy may still be susceptible to a low reward

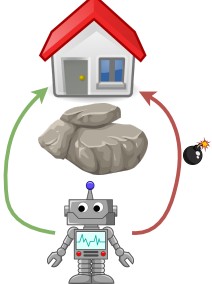

**Figure 1:** Policies have different vulnerabilities.

under attacks, as a very small divergence can lead it to the bomb. On the contrary, the green policy is more robust to adversarial attacks since it stays away from the bomb. Therefore, besides promoting the robustness of DNN approximators (such as the policy network), it is also important to learn a policy with stronger intrinsic robustness.

---

[*]Equal contribution.

36th Conference on Neural Information Processing Systems (NeurIPS 2022).

There is another line of work considering the long-term robustness of a deep policy under strong adversarial attacks. In particular, it is theoretically proved [54, 42] that the strongest (worst-case) attacker against a policy can be learned as an RL problem, and training the agent under such a learned attacker can result in a robust policy. Zhang et al. [52] propose the *Alternating Training with Learned Adversaries (ATLA)* framework, which alternately trains an RL agent and an RL attacker. Sun et al. [42] further propose PA-ATLA, which alternately trains an agent and the proposed more efficient PA-AD RL attacker, obtaining state-of-the-art robustness in many MuJoCo environments. However, training an RL attacker requires extra samples from the environment, and the attacker's RL problem may even be more difficult and sample expensive to solve than the agent's original RL problem [52, 42], especially in large-scale environments such as Atari games with pixel observations. Therefore, although ATLA and PA-ATLA are able to achieve high long-term reward under attacks, they double the computational burden and sample complexity to train the robust agent.

The above analysis of existing literature suggests two main challenges in improving the adversarial robustness of DRL agents: (1) correctly characterizing the long-term reward vulnerability of an RL policy, and (2) efficiently training a robust agent without requiring much more effort than vanilla training. To tackle these challenges, in this paper, we propose a generic and efficient robust training framework named *Worst-case-aware Robust RL (WocaR-RL)* that estimates and improves the long-term robustness of an RL agent.

WocaR-RL has 3 key mechanisms. First, WocaR-RL introduces a novel *worst-attack Bellman operator* which uses existing off-policy samples to estimate the lower bound of the policy value under the worst-case attack. Compared to prior works [52, 42] which attempt to learn the worst-case attack by RL methods, WocaR-RL does not require any extra interaction with the environment. Second, using the estimated worst-case policy value, WocaR-RL optimizes the policy to select actions that not only achieve high natural future reward, but also achieve high worst-case reward when there are adversarial attacks. Therefore, WocaR-RL learns a policy with less intrinsic vulnerability. Third, WocaR-RL regularizes the policy network with a carefully designed state importance weight. As a result, the DNN approximator tolerates state perturbations, especially for more important states where decisions are crucial for future reward. The above 3 mechanisms can also be interpreted from a geometric perspective of adversarial policy learning, as detailed in Appendix B.

Our **contributions** can be summarized as below. **(1)** We provide an approach to estimate the worst-case value of any policy under any bounded $\ell_p$ adversarial attacks. This helps evaluate the robustness of a policy without learning an attacker which requires extra samples and exploration. **(2)** We propose a novel and principled robust training framework for RL, named *Worst-case-aware Robust RL (WocaR-RL)*, which characterizes and improves the worst-case robustness of an agent. WocaR-RL can be used to robustify existing DRL algorithms (e.g. PPO [39], DQN [32]). **(3)** We show by experiments that WocaR-RL achieve **improved robustness** against various adversarial attacks as well as **higher efficiency**, compared with state-of-the-art (SOTA) robust RL methods in many MuJoCo and Atari games. For example, compared to the SOTA algorithm PA-ATLA-PPO [42] in the Walker environment, we obtain 20% more worst-case reward (under the strongest attack algorithm), with only about 50% training samples and 50% running time. Moreover, WocaR-RL learns **more interpretable "robust behaviors"** than PA-ATLA-PPO in Walker as shown in Figure 2.

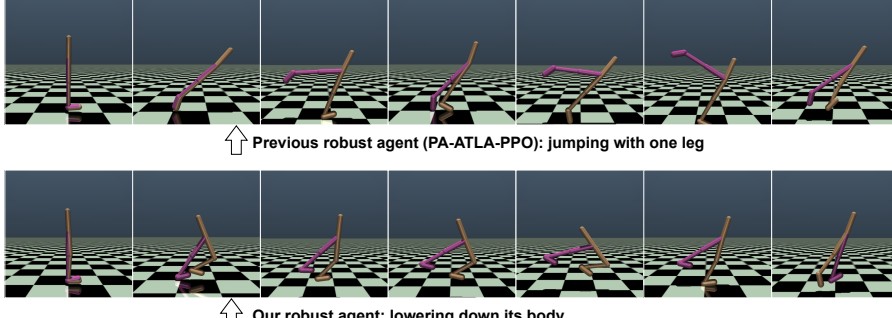

**Figure 2:** The robust Walker agents trained with **(top)** the state-of-the-art method PA-ATLA-PPO [42] and **(bottom)** our WocaR-RL. Although PA-ATLA-PPO agent also achieves high reward under attacks, it learns to jump with one leg, which is counter-intuitive and may indicate some level of overfitting to a specific attacker. In contrast, our WocaR-RL agent learns to lower down its body, which is more intuitive and interpretable. The full agent trajectories in Walker and other environments are provided in supplementary materials as GIF figures.

## 2 Related Work

**Defending against Adversarial Perturbations on State Observations.** **(1)** *Regularization-based methods* [54, 40, 33] enforce the policy to have similar outputs under similar inputs, which achieves certifiable performance for DQN in some Atari games. But in continuous control tasks, these methods may not reliably improve the worst-case performance. A recent work by Korkmaz [21] points out that these adversarially trained models may still be sensible to new perturbations. **(2)** *Attack-driven methods* train DRL agents with adversarial examples. Some early works [22, 4, 29, 34] apply weak or strong gradient-based attacks on state observations to train RL agents against adversarial perturbations. Zhang et al. [52] propose Alternating Training with Learned Adversaries (ATLA), which alternately trains an RL agent and an RL adversary and significantly improves the policy robustness in continuous control games. Sun et al. [42] further extend this framework to PA-ATLA with their proposed more advanced RL attacker PA-AD. Although ATLA and PA-ATLA achieve strong empirical robustness, they require training an extra RL adversary that can be computationally and sample expensive. **(3)** There is another line of work studying *certifiable robustness* of RL policies. Several works [27, 33, 9] computed lower bounds of the action value network $Q^\pi$ to certify robustness of action selection at every step. However, these bounds do not consider the distribution shifts caused by attacks, so some actions that appear safe for now can lead to extremely vulnerable future states and low long-term reward under future attacks. Moreover, these methods cannot apply to continuous action spaces. Kumar et al. and Wu et al.[23, 49] both extend randomized smoothing [7] to derive robustness certificates for trained policies. But these works mostly focus on theoretical analysis, and effective robust training approaches rather than robust training.

**Adversarial Defenses against Other Adversarial Attacks.** Besides observation perturbations, attacks can happen in many other scenarios. For example, the agent's executed actions can be perturbed [50, 44, 45, 24]. Moreover, in a multi-agent game, an agent's behavior can create adversarial perturbations to a victim agent [13]. Pinto et al. [35] model the competition between the agent and the attacker as a zero-sum two-player game, and train the agent under a learned attacker to tolerate both environment shifts and adversarial disturbances. We point out that although we mainly consider state adversaries, our WocaR-RL can be extended to action attacks as formulated in Appendix C.5. Note that we focus on robustness against test-time attacks, different from poisoning attacks which alter the RL training process [3, 20, 41, 56, 36].

**Safe RL and Risk-sensitive RL.** There are several lines of work that study RL under safety/risk constraints [18, 11, 10, 2, 46] or under intrinsic uncertainty of environment dynamics [26, 30]. However, these works do not deal with adversarial attacks, which can be adaptive to the learned policy. More comparison between these methods and our proposed method is discussed in Section 4.

## 3 Preliminaries and Background

**Reinforcement Learning (RL).** An RL environment is modeled by a Markov Decision Process (MDP), denoted by a tuple $\mathcal{M} = \langle \mathcal{S}, \mathcal{A}, P, R, \gamma \rangle$, where $\mathcal{S}$ is a state space, $\mathcal{A}$ is an action space, $P : \mathcal{S} \times \mathcal{A} \to \Delta(\mathcal{S})$ is a stochastic dynamics model[2], $R : \mathcal{S} \times \mathcal{A} \to \mathbb{R}$ is a reward function and $\gamma \in [0, 1)$ is a discount factor. An agent takes actions based on a policy $\pi : \mathcal{S} \to \Delta(\mathcal{A})$. For any policy, its *natural performance* can be measured by the value function $V^\pi(s) := \mathbb{E}_{P,\pi}[\sum_{t=0}^{\infty} \gamma^t R(s_t, a_t) \mid s_0 = s]$, and the action value function $Q^\pi(s, a) := \mathbb{E}_{P,\pi}[\sum_{t=0}^{\infty} \gamma^t R(s_t, a_t) \mid s_0 = s, a_0 = a]$. We call $V^\pi$ the *natural value* and $Q^\pi$ the *natural action value* in contrast to the values under attacks, as will be introduced in Section 4.

**Deep Reinforcement Learning (DRL).** In large-scale problems, a policy can be parameterized by a neural network. For example, value-based RL methods (e.g. DQN [32]) usually fit a Q network and take the greedy policy $\pi(s) = \operatorname{argmax}_a Q(s, a)$. In actor-critic methods (e.g. PPO [39]), the learner directly learns a policy network and a critic network. In practice, an agent usually follows a stochastic policy during training that enables exploration, and executes a trained policy deterministically in test-time, e.g. the greedy policy learned with DQN. Throughout this paper, we use $\pi_\theta$ to denote the training-time stochastic policy parameterized by $\theta$, while $\pi$ denotes the trained deterministic policy that maps a state to an action.

**Test-time Adversarial Attacks.** After training, the agent is deployed into the environment and executes a pre-trained fixed policy $\pi$. An attacker/adversary, during the deployment of the agent, may

---

[2] $\Delta(\mathcal{X})$ denotes the space of probability distributions over $\mathcal{X}$.

perturb the state observation of the agent/victim at every time step with a certain attack budget $\epsilon$. Note that the attacker only perturbs the inputs to the policy, and the underlying state in the environment does not change. This is a realistic setting because real-world observations can come from noisy sensors or be manipulated by malicious attacks. For example, an auto-driving car receives sensory observations; an attacker may add imperceptible noise to the camera, or perturb the GPS signal, although the underlying environment (the road) remains unchanged. In this paper, we consider the $\ell_p$ *thread model* which is widely used in adversarial learning literature: at step $t$, the attacker alters the observation $s_t$ into $\tilde{s}_t \in \mathcal{B}_\epsilon(s_t)$, where $\mathcal{B}_\epsilon(s_t)$ is a $\ell_p$ norm ball centered at $s_t$ with radius $\epsilon$. The above setting ($\ell_p$ constrained observation attack) is the same with many prior works [19, 34, 54, 52, 42].

## 4  Worst-case-aware Robust RL

In this section, we present *Worst-case-aware Robust RL (WocaR-RL)*, a generic framework that can be fused with any DRL approach to improve the adversarial robustness of an agent. We will introduce the three key mechanisms in WocaR-RL: worst-attack value estimation, worst-case-aware policy optimization, and value-enhanced state regularization, respectively. Then, we will illustrate how to incorporate these mechanisms into existing DRL algorithms to improve their robustness.

**Mechanism 1: Worst-attack Value Estimation**

Traditional RL aims to learn a policy with the maximal value $V^\pi$. However, in a real-world problem where observations can be noisy or even adversarially perturbed, it is not enough to only consider the natural value $V^\pi$ and $Q^\pi$. As motivated in Figure 1, two policies with similar natural rewards can get totally different rewards under attacks. To comprehensively evaluate how good a policy is in an adversarial scenario and to improve its robustness, we should be aware of the lowest possible long-term reward of the policy when its observation is adversarially perturbed with a certain attack budget $\epsilon$ at every step (with an $\ell_p$ attack model introduced in Section 3).

The worst-case value of a policy is, by definition, the cumulative reward obtained under the optimal attacker. As justified by prior works [54, 42], for any given victim policy $\pi$ and attack budget $\epsilon > 0$, there exists an optimal attacker, and finding the optimal attacker is equivalent to learning the optimal policy in another MDP. We denote the optimal (deterministic) attacker's policy as $h^*$. However, learning such an optimal attacker by RL algorithms requires extra interaction samples from the environment, due to the unknown dynamics. Moreover, learning the attacker by RL can be hard and expensive, especially when the state observation space is high-dimensional.

Instead of explicitly learning the optimal attacker with a large amount of samples, we propose to directly estimate the worst-case cumulative reward of the policy by characterizing the vulnerability of the given policy. We first define the *worst-attack action value* of policy $\pi$ as $\underline{Q}^\pi(s,a) := \mathbb{E}_P[\sum_{t=0}^\infty \gamma^t R(s_t, \pi(h^*(s_t))) \mid s_0 = s, a_0 = a]$. The *worst-attack value* $\underline{V}^\pi$ can be defined using $h^*$ in the same way, as shown in Definition A.1 in Appendix A. Then, we introduce a novel operator $\underline{\mathcal{T}}^\pi$, namely the *worst-attack Bellman operator*, defined as below.

**Definition 4.1** (Worst-attack Bellman Operator). For MDP $\mathcal{M}$, given a fixed policy $\pi$ and attack radius $\epsilon$, define the worst-attack Bellman operator $\underline{\mathcal{T}}^\pi$ as

$$(\underline{\mathcal{T}}^\pi Q)(s,a) := \mathbb{E}_{s' \sim P(s,a)}[R(s,a) + \gamma \min_{a' \in \mathcal{A}_{\text{adv}}(s',\pi)} Q(s',a')], \tag{1}$$

where $\forall s \in \mathcal{S}$, $\mathcal{A}_{\text{adv}}(s,\pi)$ is defined as

$$\mathcal{A}_{\text{adv}}(s,\pi) := \{a \in \mathcal{A} : \exists \tilde{s} \in \mathcal{B}_\epsilon(s) \text{ s.t. } \pi(\tilde{s}) = a\}. \tag{2}$$

Here $\mathcal{A}_{\text{adv}}(s',\pi)$ denotes the set of actions an adversary can mislead the victim $\pi$ into selecting by perturbing the state $s'$ into a neighboring state $\tilde{s} \in \mathcal{B}_\epsilon(s')$. This hypothetical perturbation to the *future* state $s'$ is the key for characterizing the worst-case long-term reward under attack. The following theorem associates the worst-attack Bellman operator and the worst-attack action value.

**Theorem 4.2** (Worst-attack Bellman Operator and Worst-attack Action Value). *For any given policy $\pi$, $\underline{\mathcal{T}}^\pi$ is a contraction whose fixed point is $\underline{Q}^\pi$, the worst-attack action value of $\pi$ under any $\ell_p$ observation attacks with radius $\epsilon$.*

Theorem 4.2 proved in Appendix A suggests that the lowest possible cumulative reward of a policy under bounded observation attacks can be computed by worst-attack Bellman operator. The corresponding worst-attack value $\underline{V}^\pi$ can be obtained by $\underline{V}^\pi(s) = \min_{a \in \mathcal{A}_{\text{adv}}(s,\pi)} \underline{Q}^\pi(s,a)$.

**How to Compute $\mathcal{A}_{\mathrm{adv}}$.** To obtain $\mathcal{A}_{\mathrm{adv}}(s, \pi)$, we need to identify the actions that can be the outputs of the policy $\pi$ when the input state $s$ is perturbed within $\mathcal{B}_\epsilon(s)$. This can be solved by commonly-used convex relaxation of neural networks [15, 55, 48, 53, 14], where layer-wise lower and upper bounds of the neural network are derived. That is, we calculate $\overline{\pi}$ and $\underline{\pi}$ such that $\overline{\pi}(s) \geq \pi(\hat{s}) \geq \underline{\pi}(s), \forall \hat{s} \in \mathcal{B}_\epsilon(s)$. With such a relaxation, we can obtain a superset of $\mathcal{A}_{\mathrm{adv}}$, namely $\hat{\mathcal{A}}_{\mathrm{adv}}$. Then, the fixed point of Equation (1) with $\mathcal{A}_{\mathrm{adv}}$ being replaced by $\hat{\mathcal{A}}_{\mathrm{adv}}$ becomes a lower bound of the worst-attack action value. For a continuous action space, $\hat{\mathcal{A}}_{\mathrm{adv}}(s, \pi)$ contains actions bounded by $\overline{\pi}(s)$ and $\underline{\pi}(s)$. For a discrete action space, we can first compute the maximal and minimal probabilities of taking each action, and derive the set of actions that are likely to be selected. The computation of $\hat{\mathcal{A}}_{\mathrm{adv}}$ is not expensive, as there are many efficient convex relaxation methods [31, 53] which compute $\overline{\pi}$ and $\underline{\pi}$ with only constant-factor more computations than directly computing $\pi(s)$. Experiment in Section 5 verifies the efficiency of our approach, where we use a well-developed toolbox auto_LiRPA [51] to calculate the convex relaxation. More implementation details and explanations are provided in Appendix C.1.

**Estimating Worst-attack Value.** Note that the worst-attack Bellman operator $\underline{\mathcal{T}}^\pi$ is similar to the optimal Bellman operator $\mathcal{T}^*$, although it uses $\min_{a \in \mathcal{A}_{\mathrm{adv}}}$ instead of $\max_{a \in \mathcal{A}}$. Therefore, once we identify $\mathcal{A}_{\mathrm{adv}}$ as introduced above, it is straightforward to compute the worst-attack action value using Bellman backups. To model the worst-attack action value, we train a network named *worst-attack critic*, denoted by $\underline{Q}^\pi_\phi$, where $\phi$ is the parameterization. Concretely, for any mini-batch $\{s_t, a_t, r_t, s_{t+1}\}^N_{t=1}$, $\underline{Q}^\pi_\phi$ is optimized by minimizing the following estimation loss:

$$\mathcal{L}_{\mathrm{est}}(\underline{Q}^\pi_\phi) := \frac{1}{N} \sum_{t=1}^{N} (\underline{y}_t - \underline{Q}^\pi_\phi(s_t, a_t))^2, \text{where } \underline{y}_t = r_t + \gamma \min_{\hat{a} \in \mathcal{A}_{\mathrm{adv}}(s_{t+1}, \pi)} \underline{Q}^\pi_\phi(s_{t+1}, \hat{a}). \quad (3)$$

For a discrete action space, $\mathcal{A}_{\mathrm{adv}}$ is a discrete set and solving $\underline{y}_t$ is straightforward. For a continuous action space, we use gradient descent to approximately find the minimizer $\hat{a}$. Since $\mathcal{A}_{\mathrm{adv}}$ is in general small, this minimization is usually easy to solve. In MuJoCo, we find that 50-step gradient descent already converges to a good solution with little computational cost, as detailed in Appendix D.3.3.

**Differences with Worst-case Value Estimation in Related Work.** Our proposed worst-attack Bellman operator is different from the worst-case Bellman operator in the literature of risk-sensitive RL [18, 11, 43, 10, 2, 46], whose goal is to avoid unsafe trajectories under the intrinsic uncertainties of the MDP. These inherent uncertainties of the environment are independent of the learned policy. In contrast, our focus is to defend against adversarial perturbations created by malicious attackers that can be *adaptive* to the policy. The GWC reward proposed by [33] also estimates the worst-case reward of a policy under state perturbations. But their evaluation is based on a greedy strategy and requires interactions with the environment, which is different from our estimation.

**Mechanism 2: Worst-case-aware Policy Optimization**

So far we have introduced how to evaluate the worst-attack value of a policy by learning a worst-attack critic. Inspired by the actor-critic framework, where the actor policy network $\pi_\theta$ is optimized towards a direction that the critic value increases the most, we can regard worst-attack critic as a special critic that directs the actor to increase the worst-attack value. That is, we encourage the agent to select an action with a higher worst-attack action value, by minimizing the worst-attack policy loss below:

$$\mathcal{L}_{\mathrm{wst}}(\pi_\theta; \underline{Q}^\pi_\phi) := -\frac{1}{N} \sum_{t=1}^{N} \sum_{a \in \mathcal{A}} \pi_\theta(a|s_t) \underline{Q}^\pi_\phi(s_t, a), \quad (4)$$

where $\underline{Q}^\pi_\phi$ is the worst-attack critic learned via $\mathcal{L}_{\mathrm{est}}$ introduced in Equation (3). Note that $\mathcal{L}_{\mathrm{wst}}$ is a general form, while the detailed implementation of the worst-attack policy optimization can vary depending on the architecture of $\pi_\theta$ in the base RL algorithm (e.g. PPO has a policy network, while DQN acts using the greedy policy induced by a Q network). In Appendix C.2 and Appendix C.3, we illustrate how to implement $\mathcal{L}_{\mathrm{wst}}$ for PPO and DQN as two examples.

The proposed worst-case-aware policy optimization has several **merits** compared to prior ATLA [52] and PA-ATLA [42] methods which alternately train the agent and an RL attacker. **(1)** Learning the optimal attacker $h^*$ requires collecting extra samples using the current policy (on-policy estimation). In contrast, $\underline{Q}^\pi_\phi$ can be learned using off-policy samples, e.g., historical samples in the replay buffer, and thus is more suitable for training where the policy changes over time. ($\underline{Q}^\pi_\phi$ depends on the current

policy via the computation of $\mathcal{A}_{\mathrm{adv}}$.) **(2)** We properly exploit the policy function that is being trained by computing the set of possibly selected actions $\hat{\mathcal{A}}_{\mathrm{adv}}$ for any state. In contrast, ATLA [52] learns an attacker by treating the current policy as a black box, ignoring the intrinsic properties of the policy. PA-ATLA [42], although assumes white-box access to the victim policy, also needs to explore and learn from extra on-policy interactions. **(3)** The attacker trained with DRL methods, namely $\hat{h}^*$, is not guaranteed to converge to an optimal solution, such that the performance of $\pi$ estimated under $\hat{h}^*$ can be overly optimistic. Our estimation, as mentioned in Mechanism 1, computes a lower bound of $\underline{Q}^\pi$ and thus can better indicate the robustness of a policy.

**Mechanism 3: Value-enhanced State Regularization**

As discussed in Section 1, the vulnerability of a deep policy comes from both the policy's intrinsic vulnerability with the RL dynamics and the DNN approximator. The first two mechanisms of WocaR-RL mainly focus on the policy's intrinsic vulnerability, i.e., let the policy select actions that are less vulnerable to possible attacks in all future steps. However, if a bounded state perturbation can cause the network to output a very different action, then the $\mathcal{A}_{\mathrm{adv}}$ set will be large and $\underline{Q}^\pi$ can thus be low. Therefore, it is also important to encourage the trained policy to output similar actions for the clean state $s$ and any $\tilde{s} \in \mathcal{B}_\epsilon(s)$, as is done in prior work [54, 40, 9].

But different from these prior methods, we note that different states should be treated differently. Some states are "critical" where selecting a bad action will result in catastrophic consequences. For example, when the agent gets close to the bomb in Figure 1, we should make the network more resistant to adversarial state perturbations. To differentiate states based on their impacts on future reward, we propose to measure the importance of states with Definition 4.3 below.

**Definition 4.3** (State Importance Weight). Define state importance weight of $s \in \mathcal{S}$ for policy $\pi$ as
$$w(s) = \max_{a_1 \in \mathcal{A}} Q^\pi(s, a_1) - \min_{a_2 \in \mathcal{A}} Q^\pi(s, a_2). \tag{5}$$

To justify whether Definition 4.3 can characterize state importance, we train a DQN network in an Atari game Pong, and show the states with the highest weight and the lowest weight in Figure 3, among many state samples. We can see that the state with higher weight in Figure 3(left) is indeed crucial for the game, as the green agent paddle is close to the ball. Conversely, a less-important state in Figure 3(right) does not have significantly different future rewards under different actions. Computing $w(s)$ is easy in a discrete action space, while in a continuous action space, one can use gradient descent to approximately find the maximal and the minimal Q values for a state. Similar to the

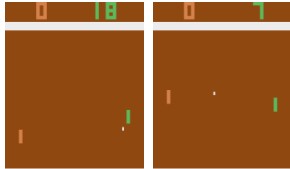

**Figure 3:** States in Pong with **(left)** high weight $w(s)$ and **(right)** low weight $w(s)$.

computation of Equation (3) with a continuous action space, we find that a 50-step gradient descent works well in experiments.

By incorporating the state importance weight $w(s)$, we regularize the policy network and let it pay more attention to more crucial states, by minimizing the following loss:
$$\mathcal{L}_{\mathrm{reg}}(\pi_\theta) = \frac{1}{N} \sum_{t=1}^{N} w(s_t) \max_{\tilde{s}_t \in \mathcal{B}_\epsilon(s_t)} \mathsf{Dist}(\pi_\theta(s_t), \pi_\theta(\tilde{s}_t)), \tag{6}$$

where Dist can be any distance measure between two distributions (e.g., KL-divergence). Minimizing $\mathcal{L}_{\mathrm{reg}}$ can result in a smaller $\mathcal{A}_{\mathrm{adv}}$, and thus the worst-attack value will be closer to the natural value.

**WocaR-RL: A Generic Robust Training Framework**

So far we have introduced three key mechanisms and their loss functions, $\mathcal{L}_{\mathrm{est}}$ in Equation (3), $\mathcal{L}_{\mathrm{wst}}$ in Equation (4) and $\mathcal{L}_{\mathrm{reg}}$ in Equation (6). Then, our robust training framework WocaR-RL combines these losses with any base RL algorithm. To be more specific, as shown in Figure 4, for any base RL algorithm that trains policy $\pi_\theta$ using loss $\mathcal{L}_{\mathrm{RL}}$, we learn an extra worst-attack critic network $\underline{Q}^\pi_\phi$ by minimizing

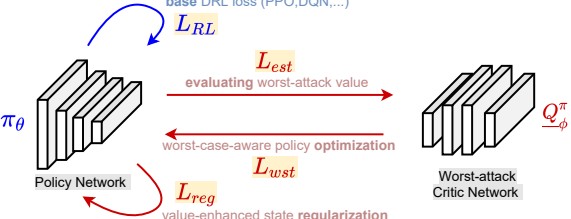

**Figure 4:** Training architecture of WocaR-RL. (Components proposed in this paper are colored as red.)

$$\mathcal{L}_{\underline{Q}^\pi_\phi} := \mathcal{L}_{\mathrm{est}}(\underline{Q}^\pi_\phi), \tag{7}$$

and combine $\mathcal{L}_{\mathrm{wst}}$ and $\mathcal{L}_{\mathrm{reg}}$ with $\mathcal{L}_{\mathrm{RL}}$ to optimize $\pi_\theta$ by minimizing

$$\mathcal{L}_{\pi_\theta} := \mathcal{L}_{\mathrm{RL}}(\pi_\theta) + \kappa_{\mathrm{wst}}\mathcal{L}_{\mathrm{wst}}(\pi_\theta; \underline{Q}_\phi^\pi) + \kappa_{\mathrm{reg}}\mathcal{L}_{\mathrm{reg}}(\pi_\theta), \tag{8}$$

where $\kappa_{\mathrm{wst}}$ and $\kappa_{\mathrm{reg}}$ are hyperparameters balancing between natural performance and robustness. Note that $\underline{Q}_\phi^\pi$ is trained together but independently with $\pi_\theta$ using historical transition samples, so WocaR-RL does not require extra samples from the environment. WocaR-RL can also be interpreted from a geometric perspective based on prior RL polytope theory [8, 42] as detailed in Appendix B.

Our WocaR-RL is a generic robust training framework that can be used to robustify existing DRL algorithms. We provide two case studies: **(1)** combining WocaR-RL with a policy-based algorithm PPO [39], namely *WocaR-PPO*, and **(2)** combining WocaR-RL with a value-based algorithm DQN [32], namely *WocaR-DQN*. The pseudocodes of WocaR-PPO and WocaR-DQN are illustrated in Appendix C.2 and Appendix C.3. The application of WocaR-RL to other DRL methods is then straightforward, since most DRL methods are either policy-based or value-based. Next, we show by experiments that WocaR-PPO and WocaR-DQN achieve state-of-the-art robustness with superior efficiency, in various continuous control tasks and video game environments. We also empirically verify the effectiveness of each of the 3 mechanisms of WocaR-RL and their weights by ablation study in Section 5.2.

## 5 Experiments and Discussion

In this section, our experimental evaluations on various MuJoCo and Atari environments aim to study the following questions: **(1)** Can WocaR-RL learn policies with better **robustness** under existing strong adversarial attacks? **(2)** Can WocaR-RL maintain **natural performance** when improving robustness? **(3)** Can WocaR-RL learn more **efficiently** during robust training? **(4)** Is each mechanism in WocaR-RL **effective**? Problem (1), (2) and (3) are answered in Section 5.1 with detailed empirical results, and problem (4) is studied in Section 5.2 via ablation experiments.

### 5.1 Experiments and Evaluations

**Environments.**    Following most prior works [54, 52, 33] and the released implementation, we apply our WocaR-RL to PPO [39] on 4 MuJoCo tasks with continuous action spaces, including Hopper, Walker2d, Halfcheetah and Ant, and to DQN [32] agents on 4 Atari games including Pong, Freeway, BankHeist and RoadRunner, which have high dimensional pixel inputs and discrete action spaces.

**Baselines and Implementation.**    We compare our algorithm with several state-of-the-art robust training methods, including (1) *SA-PPO/SA-DQN* [54]: regularizing policy networks by convex relaxation. (2) *ATLA-PPO* [52]: alternately training an agent and an RL attacker. (3) *PA-ATLA-PPO* [42]: alternately training an agent and a more advanced RL attacker PA-AD. (4) *RADIAL-PPO/RADIAL-DQN* [33]: optimizing policy network by designed adversarial loss functions based on robustness bounds. SA and RADIAL have both PPO and DQN versions, which are compared with our WocaR-PPO and WocaR-DQN. But ATLA and PA-ATLA do not provide DQN versions, since alternately training on DQN can be expensive as explained in the original papers [42]. (PA-ATLA has an A2C version, which we compare in Appendix D.2.) Therefore, we reproduce their ATLA-PPO and PA-ATLA-PPO results and compare them with our WocaR-PPO. More implementation and hyperparameter details are provided in Appendix D.1.

#### Case I: Robust PPO for MuJoCo Continuous Control

**Evaluation Metrics.**    To reflect both the natural performance and robustness of trained agents, we report the average episodic rewards under no attack and against various attacks. For a comprehensive robustness evaluation, we attack the trained robust models with multiple existing attack methods, including: (1) *MaxDiff* [54] (maximal action difference), (2) *Robust Sarsa (RS)* [54] (attacking with a robust action-value function), (3) *SA-RL* [54] (finding the optimal state adversary) and (4) *PA-AD* [42] (the *existing strongest attack* by learning the optimal policy adversary with RL). For a clear comparison, we use the same attack radius $\epsilon$ as in most baselines [54, 52, 42].

**Performance and Robustness of WocaR-PPO**    Figure 5 (left four columns) shows performance curves during training under four different adversarial attacks. Among all four attack algorithms, WocaR-PPO converges much faster than baselines, and often achieves the best asymptotic robust performance, especially under the strongest PA-AD attack. It is worth emphasizing that since we train a robust agent without explicitly learning an RL attacker, our method not only obtains

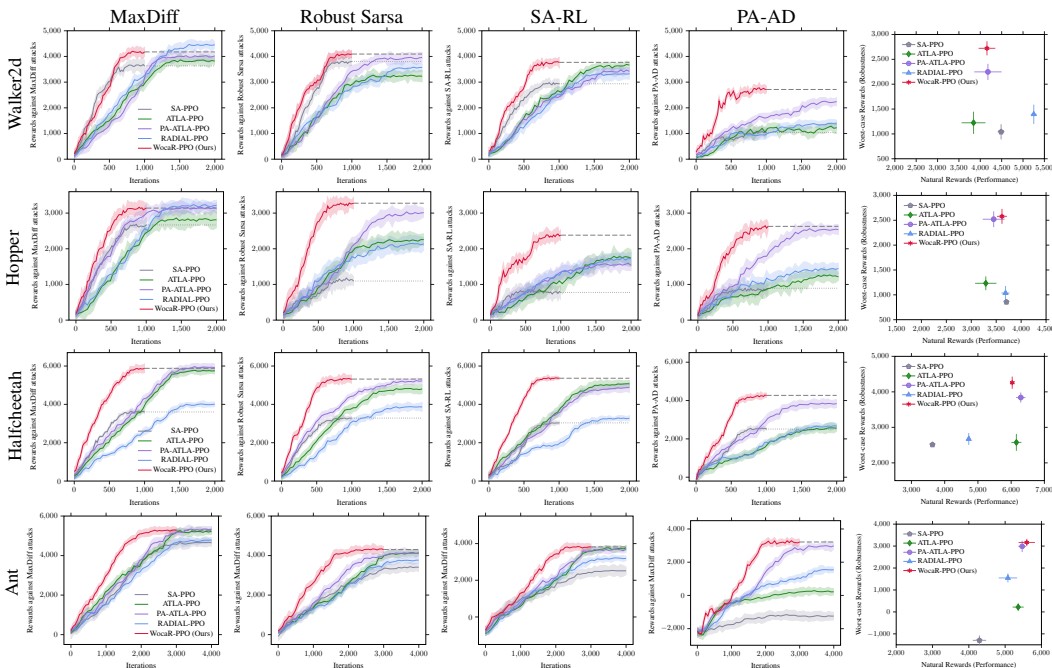

**Figure 5: Robustness, Efficiency and High Natural Performance of WocaR-PPO. (Left four columns)** Learning curves of rewards under MaxDiff, Robust Sarsa, SA-RL and PA-AD *(the strongest)* attacks during training on four environments. **(Rightmost column)** Average episode natural rewards v.s. average worst rewards under attacks. Each row shows the performance of baselines and WocaR-PPO on one environment. Shaded regions are computed over 20 random seeds. Results under more attack radius $\epsilon$'s are in Appendix D.3.1.

stronger robustness and much higher efficiency, but also a more general defense: WocaR-PPO obtains comprehensively superior performance against a variety of attacks compared against existing SOTA algorithms based on learned attackers (ATLA-PPO, PA-ATLA-PPO). Additionally, in our experiments, WocaR-PPO learns relatively more universal defensive behaviors as shown in Figure 2, which can physically explain why our algorithm can defend against diverse attacks. We provide policy demonstrations in multiple tasks in our supplementary materials.

The comparison of natural performance and the worst-case performance appears in Figure 5 (right). We see that WocaR-PPO maintains competitive natural rewards under no attack compared with other baselines, which demonstrates that our algorithm gains more robustness without losing too much natural performance. The full results of baselines and our algorithm under different attack evaluations are provided by Table 2 in Appendix D.2 (including performance under random attacks).

**Efficiency of Training WocaR-PPO.** The learning curves in Figure 5 (left) directly show the sample efficiency of WocaR-PPO. Following the optimal settings provided in [54, 52, 33], our method takes *50% training steps* required by RADIAL-PPO and ATLA methods on Hopper, Walker2d, and Halfcheetah because RADIAL-PPO needs more steps to ensure convergence and ATLA methods require additional adversary training steps. When solving high dimensional environments like Ant, WocaR-PPO only requires *75% steps* compared with all other baselines to converge. We also provide additional results of baselines using the same training steps as WocaR-PPO in Appendix D.3.2.

In terms of time efficiency, WocaR-PPO saves *50% training time* for convergence on Hopper, Walker2d, and Halfcheetah, and *32% time* on Ant compared with the SOTA method. Therefore, *WocaR-PPO achieves both higher computational efficiency and higher sample efficiency than SOTA baselines.* Detailed costs in time and sampling are in Appendix D.3.3.

**Case II: Robust DQN for Atari Video Games**

**Evaluation Metrics.** Since Atari games have pixel state spaces and discrete action spaces, the applicable attacking algorithms also differ from those in MuJoCo tasks. We include the following common attacks: (1) 10-step untargeted *PGD* (projected gradient descent) attack, (2) *MinBest* [19], which minimizes the probability of choosing the "best" action, (3) *PA-AD* [42], as the state-of-the-art RL-based adversarial attack algorithm.

| Model | Pong | | | | BankHeist | | | |
|---|---|---|---|---|---|---|---|---|
| | Natural Reward | PGD | MinBest | PA-AD | Natural Reward | PGD | MinBest | PA-AD |
| | | | $\epsilon = 3/255$ | | | | $\epsilon = 3/255$ | |
| DQN | $21.0 \pm 0.0$ | $-21.0 \pm 0.0$ | $-9.7 \pm 4.0$ | $-19.0 \pm 2.2$ | $\mathbf{1308 \pm 24}$ | $0 \pm 0$ | $119 \pm 65$ | $102 \pm 92$ |
| SA-DQN | $21.0 \pm 0.0$ | $21.0 \pm 0.0$ | $20.6 \pm 3.5$ | $18.7 \pm 2.6$ | $1245 \pm 14$ | $1176 \pm 63$ | $1024 \pm 31$ | $489 \pm 106$ |
| RADIAL-DQN | $21.0 \pm 0.0$ | $21.0 \pm 0.0$ | $19.5 \pm 2.1$ | $13.2 \pm 1.8$ | $1178 \pm 4$ | $1176 \pm 63$ | $928 \pm 113$ | $508 \pm 85$ |
| **WocaR-DQN (Ours)** | $\mathbf{21.0 \pm 0.0}$ | $\mathbf{21.0 \pm 0.0}$ | $\mathbf{20.8 \pm 3.3}$ | $\mathbf{19.7 \pm 2.4}$ | $1220 \pm 12$ | $\mathbf{1214 \pm 7}$ | $\mathbf{1045 \pm 20}$ | $\mathbf{754 \pm 102}$ |

| Model | Freeway | | | | RoadRunner | | | |
|---|---|---|---|---|---|---|---|---|
| | Natural Reward | PGD | MinBest | PA-AD | Natural Reward | PGD | MinBest | PA-AD |
| | | | $\epsilon = 3/255$ | | | | $\epsilon = 3/255$ | |
| DQN | $\mathbf{34.0 \pm 0.1}$ | $0.0 \pm 0.0$ | $5.5 \pm 1.8$ | $4.7 \pm 2.9$ | $\mathbf{45527 \pm 4894}$ | $0 \pm 0$ | $2985 \pm 1440$ | $203 \pm 65$ |
| SA-DQN | $30.0 \pm 0.0$ | $30.0 \pm 0.0$ | $18.3 \pm 3.0$ | $9.5 \pm 3.8$ | $44638 \pm 2367$ | $20678 \pm 1563$ | $4214 \pm 2587$ | $5516 \pm 4684$ |
| RADIAL-DQN | $33.1 \pm 0.2$ | $\mathbf{33.2 \pm 0.2}$ | $16.4 \pm 2.3$ | $10.8 \pm 3.6$ | $44675 \pm 5854$ | $38576 \pm 1960$ | $8476 \pm 3964$ | $1290 \pm 4015$ |
| **WocaR-DQN (Ours)** | $31.2 \pm 0.4$ | $31.4 \pm 0.3$ | $\mathbf{19.8 \pm 3.8}$ | $\mathbf{12.3 \pm 3.2}$ | $44156 \pm 2279$ | $\mathbf{38720 \pm 1765}$ | $\mathbf{10545 \pm 2984}$ | $\mathbf{8239 \pm 2766}$ |

**Table 1: Robustness and High Natural Performance of WocaR-DQN.** Average episode rewards $\pm$ standard deviation over 50 episodes on three baselines and WocaR-DQN on four Atari environments. Best results (natural reward of under attacks for each column) on each environment boldfaced. WocaR-DQN outperforms all the baselines in most cases or gains similar performance in the other metrics. We highlight the most robust agent as gray . Each result is obtained with 10 random seeds.

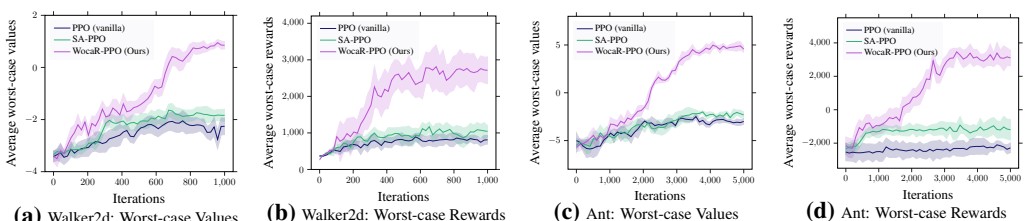

**(a)** Walker2d: Worst-case Values  **(b)** Walker2d: Worst-case Rewards  **(c)** Ant: Worst-case Values  **(d)** Ant: Worst-case Rewards

**Figure 6: (a)&(b)** Comparison between estimated worst-attack action values $\underline{Q}_\phi^\pi$ and Actual worst-case rewards under the strongest attacksduring training on Walker2d; **(c)&(d)** The comparison between worst-case values and rewards to verify worst-attack value estimation on Ant.

**Performance and Robustness of WocaR-DQN.** Table 1 presents the results on four Atari games under attack radius $\epsilon = 3/255$, while results and analysis under smaller attack radius $1/255$ are in Appendix D.2. We can see that *our WocaR-DQN consistently outperforms baselines under MinBest and PA-AD attacks in all environments, with a significant advance under the strongest (worst-case) PA-AD attacks compared with other robust agents.* Under PGD attacks, WocaR-DQN performs comparably with the state-of-the-art in Freeway and Pong (which are simpler games) and gains higher rewards than other agents in BankHeist and Roadrunner. Since SA-DQN and RADIAL-DQN focus on bounding and smoothing the policy network and do not consider the policy's intrinsic vulnerability, they are robust under the PGD attack but still vulnerable against the stronger PA-AD attack.

**Efficiency of Training WocaR-DQN.** The total training time for SA-DQN, RADIAL-DQN, and our WocaR-DQN are roughly 35, 17, and 18 hours, respectively. All baselines are trained for 6 million frames on the same hardware. Therefore, WocaR-DQN is 49% faster (and is more robust) than SA-DQN. Compared to the more advanced baseline RADIAL-DQN, although WocaR-DQN is 5% slower, it achieves better robustness (539% higher reward than RADIAL-DQN in RoadRunner).

### 5.2 Verifying Effectiveness of WocaR-RL

Now we dive deeper into the algorithmic design and verify the effectiveness of WocaR-RL by ablation studies on WocaR-PPO.

**(1) Worst-attack value estimation.** We show the learned worst-attack value estimation, $\underline{Q}_\phi^\pi$, during the training process in Figure 6a and 6c, in comparison with the actual reward under the strongest attack (PA-AD [42]) in Figure 6b and 6d. The pink curves in both plots suggest that *our worst-attack value estimation matches the trend of actual worst-case reward under attacks,* although the network estimated value and the real reward have different scales due to the commonly-used reward normalization for learning stability. Therefore, the effectiveness of our proposed worst-attack value estimation ($\mathcal{L}_{\text{est}}$) is verified.

**(2) Worst-case-aware policy optimization.** Compared to vanilla PPO and SA-PPO, we can see that *WocaR-PPO improves the worst-attack value and the worst-case reward during training*, suggesting the effectiveness of our worst-attack value improvement ($\mathcal{L}_{\text{wst}}$). The comparison of natural rewards,

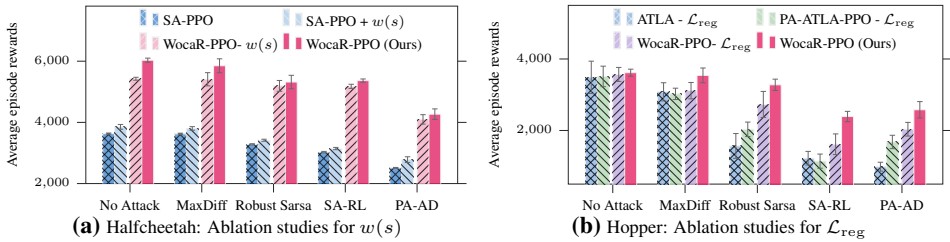

**Figure 7: (a)** Ablation evaluations for state importance weight $w(s)$ under no attack and four types of attacks on Halfcheetah; **(b)** Ablation studies for state regularization $\mathcal{L}_{\mathrm{reg}}$ under different evaluation metrics on Hopper. Ablated results on other environments are in Appendix D.3.6.

as well as curves in other environments, are provided in Appendix D.3.4. Moreover, the adjustable weight $\kappa_{\mathrm{wst}}$ in Equation (8) controls the trade-off between natural value and worst-attack value in policy optimization. When $\kappa_{\mathrm{wst}}$ is high, the policy pays more attention to its worst-attack value. Appendix D.3.5 verifies that *WocaR-RL, with different values of weight $\kappa_{\mathrm{wst}}$, produces different robustness and natural performance while consistently dominating other robust agents.*

**(3) Value-enhanced state regularization.** We conduct ablation experiments to analyze the effect of two techniques: our proposed state importance weight $w(s)$ and the state regularization loss $\mathcal{L}_{\mathrm{reg}}$ [54]. In Figure 7a, we compare the performance of the original WocaR-PPO to a variant of WocaR-PPO without the state importance weight $w(s)$ on Halfcheetah, which visually indicates that $w(s)$ can help agents boost the robustness. Since SA-PPO [54] also uses a state regularization technique, the improvement of SA-PPO added with $w(s)$ also show the universal effectiveness of our state importance. Without $w(s)$, our algorithm also achieves similar or better performance than baselines, but including this inexpensive technique $w(s)$ gives WocaR-RL a greater advantage, especially under learned strong attacks SA-RL and PA-AD. Figure 7b presents the performance of ATLA methods and our algorithm without $\mathcal{L}_{\mathrm{reg}}$ on Hopper, which verifies that WocaR-PPO also yields the superior performance when removing the regularization technique. And the comparison between WocaR-PPO and WocaR-PPO without $\mathcal{L}_{\mathrm{reg}}$ demonstrates that the weighted state regularization is beneficial to enhancing the robustness in our algorithm. Detailed ablation studies for $w(s)$ and $\mathcal{L}_{\mathrm{reg}}$ on four MuJoCo environments are shown in Appendix D.3.6.

## 6  Conclusion and Discussion

This paper proposes a robust RL training framework, WocaR-RL, that evaluates and improves the long-term robustness of a policy via worst-attack value estimation, worst-case-aware policy optimization, and value-enhanced state regularization. Different from recent state-of-the-art adversarial training methods [42, 52] which train an extra adversary to improve the robustness of an agent, we directly estimate and improve the lower bound of the agent's cumulative reward. As a result, WocaR-RL not only achieves better robustness than state-of-the-art robust RL approaches, but also halves the total sample complexity and computation complexity, in a wide range of Atari and MuJoCo tasks.

There are several aspects to improve or extend the current approach. First, the proposed worst-attack Bellman operator in theory gives the exact worst-case value of a policy under $\ell_p$ bounded attacks. But in practice, it is hard to compute the set $\mathcal{A}_{\mathrm{adv}}$ directly, so we use convex relaxation to obtain a superset of it, $\hat{\mathcal{A}}_{\mathrm{adv}}$. As a result, the fixed point of worst-attack Bellman operator with $\mathcal{A}_{\mathrm{adv}}$ being replaced by $\hat{\mathcal{A}}_{\mathrm{adv}}$ is a lower bound of the worst-case value. Then, our algorithm increases the worst-case value by improving its lower bound, as visualized and explained in Figure 8 in Appendix B. Therefore, one potential way of further improving the robustness is using a tighter relaxation. In addition, this paper only considers the $\ell_p$ threat model as is common in most related works. But in real-world applications, other attack models could exist (e.g. patch attacks [5]), and improving the robustness of RL agents in these scenarios is another important research direction.

## Acknowledgments

This work is supported by DOD-ONR-Office of Naval Research, DOD-DARPA-Defense Advanced Research Projects Agency Guaranteeing AI Robustness against Deception (GARD), and Adobe, Capital One and JP Morgan faculty fellowships.

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
