# OpenReview forum: "Efficient Adversarial Training without Attacking: Worst-Case-Aware Robust Reinforcement Learning"
_NeurIPS.cc/2022/Conference — NeurIPS 2022 Accept_

### Official Review · Reviewer_J3o4 · 2022-07-10

**Rating:** 7
**Confidence:** 4
**Soundness:** 3 good
**Presentation:** 4 excellent
**Contribution:** 3 good

**Summary:**

This paper presents a new algorithm for training deep reinforcement learning agents robust against input perturbations. It is based on using certified bounds to provide attack-agnostic defense, while also combining with a learned worst case value function to incentivize agents to choose safe policies. Its results are evaluated on MuJoCo and Atari environments showing equal or better performance and efficiency as existing state of the art robust training algorithm, with especially large gains against stronger attacks.

**Questions:**

In figure 5, is attack applied at every step during training, or just used to evaluate checkpoints in "test time"? If yes, are there experiments without train-time attacks as certification based defenses like [32, 54](and WocaR if I understand correctly) don’t require train time attacks to be robust?

Weird design choices:
- Why is it necessary to train a separate clean policy Q for WocaR-DQN? This is not necessary for comparable baselines [32, 54], why is it needed now?
- Appendix Line 741-742: “For reproducibility, the reported results are selected from 30 agents for different training methods with medium performance due to the high variance in RL training.” Is this common practice? What are the results when averaging over all runs, not just median 20?

Minor errors to be fixed:

- Line 256: L_{reg} in Equation (4) → L_{wst} in Equation (4)
- Figure 5: The colors of SA-PPO and RADIAL-PPO are too similar, hard to tell which one is which
- Appendix table 2: Walker first two columns should bold RADIAL-PPO instead of SA-PPO

**Limitations:**

Limitations are not really discussed, even though authors claim to do so in Appendix B. Some limitations that could be mentioned are that their methods are restricted defending against to l_p - norm bounded attacks which may not be the most useful setting for real world use.

Discussion of potential impact:
"Our goal is to efficiently improve RL robustness, which does not have any negative impacts." - While I agree with this in principle, the wording is too strong here, instead should say something like we don't think has any negative impacts. Plausible negative effect could for example be that more robust rl algorithms speed up adoption of robots to replace some occupations, causing mass unemployment.

### Edit:

Thank you for the detailed response. While these responses have not changes my overall view of the paper, they have addressed my outstanding concerns and clarified certain parts of the paper. In response I have adjusted my score for presentation from 3->4.

**Strengths And Weaknesses:**

Strengths:
- Novel and good ideas, especially the usage of learned worst-attack critic, as well as state-importance weights.
- Worst attack critic and Worst-case aware policy optimization provide an efficient and powerful solution to the issue of defending against powerful attacker with long-term goals.
- Extensive empirical evaluation and good empirical results, especially against stronger attacks.
- Clear writing and overall good discussion of related work

Weaknesses:
- Some strange design choices (see questions for more details)
- Missing an important connection to related work: Worst Attack Value described in section 4.1(when using convex relaxation to solve A_adv) is mathematically the same as Greedy Worst Case reward evaluation metric defined in Algorithm 1 of [32] (up to discount term). This is not an issue for the novelty of the paper as it is used for very different purposes, but the connection should be acknowledged and discussed in the paper.
- Minor issues with the mathematical notation used to describe worst attack Bellman operator:
Specifically, using convex relaxation to approximate A_adv does not solve the worst attack Bellman operator as defined in Eq. 1 but instead gives a lower bound for it. While this is somewhat discussed in the Appendix, this distinction should be clarified in the main text.
- Paper could be even stronger if evaluated on more realistic settings, such as real world applicable robotics.

---

> ### Author Response · Authors · 2022-08-01
> **[2/2] Response to questions and limitations**
>
> $\newcommand{jo}{\textcolor{orange}{\mathrm{J3o4}}}$
> > Q4: In figure 5, is attack applied at every step during training, or just used to evaluate checkpoints in "test time"? If yes, are there experiments without train-time attacks as certification based defenses like [32, 54] (and WocaR if I understand correctly) don’t require train time attacks to be robust?
>
> Yes, your understanding is correct. We just evaluate each checkpoint to visualize the training process. During training, our agents never need to train attackers, which makes WocaR-RL more efficient.
>
> > Q5: Why is it necessary to train a separate clean policy Q for WocaR-DQN? This is not necessary for comparable baselines [32, 54], why is it needed now?
>
>
> Both [32] and [54] are based on the regularization of policy/Q networks, for which training the original Q network with the regularization loss is enough. Their regularization encourages the policy to output similar actions under similar state inputs for every decision. In contrast, our work estimates the worst-case values to enhance robustness, which considers not only the current step, but also future perturbations. For this purpose, we train a worst-attack critic function $\underline{Q}^\pi_{\phi}$. However, if we choose action only based on this worst-case value without the clean Q value, it can be overly conservative and may not maintain high natural performance. This is why we still learn the clean Q network.
>
> On the other hand, there exists a **trade-off between robustness and natural performance**, which has been well studied theoretically and empirically in adversarial supervised learning works [1,2,3,4], although rarely discussed in RL setting. In our experiments, we also observe this trade-off, as shown in Appendix D.3.5, which empirically proves the above motivation for the algorithm design. Motivated by this trade-off, we combine the worst-case value and the clean value to balance robustness and natural performance. This phenomenon can also be understood from a geometric perspective, as discussed in Appendix B. In Figure 7, we can see that improving the worst-case value (bottom leftmost vertex of the pink polytope) doesn't necessarily improve the clean value (top rightmost vertex of the pink polytope). It suggests that a clean Q network is still needed for improving the natural performance of the policy.
>
> In addition, although we train both the worst-case Q function and the clean one, they are updated using the same samples. Therefore, our method does not need extra sampling during training compared with ATLA methods [41,52], which ensures the efficiency of robust training.
>
>
> > Q6: Appendix Line 741-742: “For reproducibility, the reported results are selected from 30 agents for different training methods with medium performance due to the high variance in RL training.” Is this common practice? What are the results when averaging over all runs, not just median 20?
>
> Reporting the medium results is a common experimental setting used, for example in [54], [32]. Since average values are easily affected by the extreme results which may occur in RL training because of the choice of random seeds, we report medium performance for a fair comparison with baselines.
>
>
> > Q7: Minor errors in Equation (4), colors in Figure 5 and Appendix Table 2
>
> We have fixed these typos in our updated submission. Thanks for your careful and useful advice again.
>
> **About Limitations and Negative Impact**
>
> Thank you so much for your suggestions. We have added a section in Appendix E where limitations and negative impacts are listed.
>
> ---
>
> We greatly appreciate Reviewer $\jo$'s valuable feedback and constructive suggestions. We hope our answers have addressed all the questions/concerns the reviewer has. We are happy to answer any further questions.
>
>
> Paper5837 Authors
>
>
> ---
>
> References:
>
> [1] Tsipras et al. Robustness may be at odds with accuracy. NeurIPS 2019.
>
> [2] Zhang et al. Theoretically principled trade-off between robustness and accuracy. ICML 2019.
>
> [3] Yang et al. A Closer Look at Accuracy vs. Robustness. NeurIPS 2020.
>
> [4] Mehrabi et al. Fundamental Tradeoffs in Distributionally Adversarial Training. ICML 2021.
>
>
> [32] Oikarinen et al. Robust deep reinforcement learning through adversarial loss. NeurIPS 2021.
>
> [41] Sun et al. Who is the strongest enemy? towards optimal and efficient evasion attacks in deep rl. ICLR 2022.
>
> [52] Zhang et al. Robust reinforcement learning on state observations with learned optimal adversary. ICLR 2021.
>
> [54] Zhang et al. Robust deep reinforcement learning against adversarial perturbations on state observations. NeurIPS 2020.

---

> ### Author Response · Authors · 2022-08-01
> **[1/2] Response to weakness comments**
>
> $\newcommand{jo}{\textcolor{orange}{\mathrm{J3o4}}}$
> We thank Reviewer $\jo$ for the positive comment and insightful feedback. We are encouraged that Reviewer J3o4 finds our paper novel and appreciates our idea. We have updated the paper according to Reviewer J3o4's suggestions and we address Reviewer J3o4's concerns and questions below:
>
> > Q1: Missing an important connection to related work: Worst Attack Value described in section 4.1(when using convex relaxation to solve A_adv) is mathematically the same as Greedy Worst Case reward evaluation metric defined in Algorithm 1 of [32] (up to discount term).
>
> It is a good suggestion to discuss the connection between our worst-attack value and GWC reward evaluation in [32]. The GWC algorithm proposed by [32] is a nice method that also aims to evaluate the worst-case reward of a policy under $\ell_p$-bounded attacks, and convex relaxation is also used to generate a superset of possible actions. However, our proposed evaluation method is significantly different from GWC [32], not only in the discount term. The main differences are detailed below.
>
> - GWC selects the "worst" action that has the lowest Q value or action probability. However, this heuristic-based selection of **"worst" action does not rigorously estimate the true worst-case value**, because it does not consider the influence of future attacks on the current value expectation. Note that the original Q value is the expected return of the current policy $\pi$ (or the optimal policy $\pi^*$) under no attack, and it reflects the future return of the agent if it follows $\pi$ (or $\pi^*$) in future steps. However, the original Q function may not be a good indicator of future return when $\pi$ gets attacked. It is possible that $Q^\pi(s,a_1)>Q^\pi(s,a_2)$, but taking $a_1$ leads to worse future return under future attacks. (See Figure 8 in [41] for an example of MDP.) On the contrary, our worst-attack Bellman operator is independent of the original Q value, and its fixed point is exactly the worst-case value (or the lower bound of worst-case value under convex relaxation) of the current policy. This is also why our WocaR-RL trains a separate Q network $\underline{Q}^\pi_\phi$ that is different from the clean Q network.
> - GWC is an efficient algorithm, but it still estimates the worst-case reward by interacting with the environment. In contrast, we use the existing replay buffer to do off-policy Bellman backup, so **extra interaction samples are not needed by our approach**.
>
> We added discussions of the relation with GWC in Mechanism 1 in Section 4. Please let us know if anything is unclear.
>
>
> > Q2: Minor issues with the mathematical notation used to describe worst attack Bellman operator: Specifically, using convex relaxation to approximate A_adv does not solve the worst attack Bellman operator as defined in Eq. 1 but instead gives a lower bound for it. While this is somewhat discussed in the Appendix, this distinction should be clarified in the main text.
>
> Thank you for the advice. We did explain that our evaluation is a lower bound of worst-case value due to the convex relaxation in the last line of Mechanism 2. But we are sorry that this point was not made clear in Mechanism 1. We have added more discussion about this point in the description of Mechanism 1 (Section 4).
>
> > Q3: Paper could be even stronger if evaluated in more realistic settings, such as real world applicable robotics.
>
> Due to the cost and limited access to robots, we regret that we didn't have the opportunity to evaluate our robust training method on real robotics control tasks. Actually, we think it is interesting to evaluate and enhance the robustness against perturbations in real-world robotics. We hope to enrich our empirical evaluation in robotics applications in the near future.

---

### Official Review · Reviewer_NfEV · 2022-07-11

**Rating:** 7
**Confidence:** 4
**Soundness:** 3 good
**Presentation:** 3 good
**Contribution:** 3 good

**Summary:**

The paper proposes an efficient RL training framework that is robust against $\ell_p$ adversarial attacks on states. Different from the prior work that requires additional training for an RL adversary, the proposed method directly estimates the worst-case Q values without training an extra RL adversary. The paper introduces a new Bellman operator named the Worst-attack Bellman operator and computes the worst-case Q values by minimizing the Bellman error. Given the worst-attack Q-function, a policy is optimized to maximize the Q-function via standard RL algorithms such as PPO or DQN. Finally, the paper proposes a policy regularization technique that selectively minimizes the action changes under adversarial perturbations based on the state importance weight $w$. The experimental results show the effectiveness of the proposed method in MuJoCo and Atari environments.

**Questions:**

**Estimating Worst-attack Value**

To compute the target of the Bellman error, one has to solve the problem of minimizing the Q-function, which can be infeasible when the action space is continuous. The authors say that this can be solved approximately via gradient descent (line 203), but running multi-step gradient descent for every Q-learning step is computationally expensive (like adversarial training). Could you please provide more details on how to solve this minimization problem? If the authors use an additional policy network (RL adversary) to approximate the minimization problem analogous to standard actor-critic algorithms, the authors should clearly explain the difference between the proposed method and the prior work that trains an RL adversary explicitly.

**Convex relaxation technique**

I am just wondering about how the convex relaxation technique can scale to environments with high-dimensional input spaces such as Atari. In [1, 2], this technique is only tested on small networks trained on datasets with small input sizes, such as MNIST and CIFAR10. Can this technique be applied to larger networks such as ResNet used in IMPALA [3]?

If these two problems are well-addressed in the rebuttal, I am willing to raise my score.

[1] Wong and Kolter, Provable Defenses against Adversarial Examples via the Convex Outer Adversarial Polytope, ICML 2018. \
[2] Zhang, et al., Towards Stable and Efficient Training of Verifiably Robust Neural Networks, ICLR 2020. \
[3] Espeholt, et al., IMPALA: Scalable Distributed Deep-RL with Importance Weighted Actor-Learner Architectures ICML 2018.

---
Post-rebuttal: Since all of my concerns have been well addressed, I decide to raise my score from 6 to 7.

**Limitations:**

I cannot find any potential negative impact of this paper.

**Strengths And Weaknesses:**

Strengths:
- The proposed method achieves state-of-the-art performance with a large margin.
- I think introducing a new Bellman Operator is novel and the authors formalize well its definition and properties.
- The paper is well-organized and easy to read.
- The proposed framework is simple and effective, which can be easily combined with existing RL algorithms.

Weaknesses:
- I could not find any critical problem in this paper, but some minor problems should be addressed (see Questions below).

---

> ### Author Response · Authors · 2022-07-28
> **[2/2] Explanation on Scalability of Convex Relaxation**
>
> $\newcommand{nfev}{\textcolor{blue}{\mathrm{NfEV}}}$
>
> > Q3: I am just wondering about how the convex relaxation technique can scale to environments with high-dimensional input spaces such as Atari. Can this technique be applied to larger networks such as ResNet used in IMPALA [3]?
>
>
> The effectiveness of convex relaxation in Atari environments has been verified by several prior works [32,54]. Although these papers have different purposes from ours, they also use similar convex relaxation methods for bounding the network outputs under bounded state perturbations in Atari. Our implementation of the convex relaxation follows [54], which is efficient (running time is in Line 360-364 of our submitted manuscript). Our empirical results, as well as results in prior papers [32,54], suggest that convex relaxation can scale to Atari games well.
>
> To the best of our knowledge, there are many works that effectively scale up convex relaxation techniques to high-dimensional input spaces and large-scale networks. For example, [57] successfully applies convex relaxation to complicated network architectures like WideResNet, DenseNet and ResNeXt, on Tiny-ImageNet or downscaled ImageNet datasets. Based on these scalable convex relaxation techniques, we believe that our method can be applied to improve the robustness of large-scale RL models such as IMPALA. Although the evaluation in our work and most other adversarial RL papers [32,41,52,54] focuses on shallow CNN or MLP networks, studying the robustness of large RL models is an exciting future research direction.
>
>
> ---
>
> We again thank Reviewer $\nfev$ for reviewing our paper and giving suggestions. We hope our answers have addressed all the questions/concerns the reviewer has. If so, we would greatly appreciate it if Reviewer $\nfev$ could consider raising their score. Please let us know if there are more questions.
>
>
> Paper5837 Authors
>
>
> ---
>
> References:
>
> [14] Gowal et al. Scalable verified training for provably robust image classification. ICCV 2019.
>
> [32] Oikarinen et al. Robust deep reinforcement learning through adversarial loss. NeurIPS 2021.
>
> [41] Sun et al. Who is the strongest enemy? towards optimal and efficient evasion attacks in deep rl. ICLR 2022.
>
> [52] Zhang et al. Robust reinforcement learning on state observations with learned optimal adversary. ICLR 2021.
>
> [54] Zhang et al. Robust deep reinforcement learning against adversarial perturbations on state observations. NeurIPS 2020.
>
> [57] Xu et al. "Automatic perturbation analysis for scalable certified robustness and beyond. NeurIPS 2020.

---

> ### Author Response · Authors · 2022-07-28
> **[1/2] Clarification on Computation Complexity and Differences with Prior Work**
>
> $\newcommand{nfev}{\textcolor{blue}{\mathrm{NfEV}}}$
>
> We appreciate the valuable feedback of Reviewer $\nfev$. We are particularly encouraged that the reviewer finds our method novel and effective. Below we answer the questions raised by the reviewer in detail.
>
> > Q1: To compute the target of the Bellman error, one has to solve the problem of minimizing the Q-function, which can be infeasible when the action space is continuous. The authors say that this can be solved approximately via gradient descent (line 203), but running multi-step gradient descent for every Q-learning step is computationally expensive (like adversarial training). Could you please provide more details on how to solve this minimization problem?
>
> This is a nice question, and we would like to point out that solving $\min_{a\in A} Q(s,a)$ using gradient decent is not expensive. Because the action space $A$ in general is a low-dimensional vector space, computing the gradient of $a$ is usually much cheaper than computing the gradient of a high-dimensional state perturbation as in adversarial training.
>
> In our experiments in MuJoCo with continuous actions, we solve $\min_{a \in A}$ using 50-step gradient descent. The running time of this 50-step gradient descent is about **1.68 seconds** per batch with batch size 128. In total, this gradient descent computation takes 18% of the total training time, thus it is not the computation bottleneck.
>
> Please note that training robust models is usually more expensive than standard training. Table 6 in Appendix D.3.3 shows that **our method takes less training time than all evaluated robust RL baselines** in MuJoCo tasks.
>
>
> > Q2: If the authors use an additional policy network (RL adversary) to approximate the minimization problem analogous to standard actor-critic algorithms, the authors should clearly explain the difference between the proposed method and the prior work that trains an RL adversary explicitly.
>
> We first clarify that we do not use an additional policy network, and we do not train an RL adversary. Instead, we learn an additional worst-attack critic network (we think this is what the reviewer was referring to). In **Line 221-233** of our submitted main paper (Line 222-234 in revised version), we have explained **3 key differences** between our proposed method and the prior work that trains an RL adversary explicitly. In short, compared to prior works which learn an RL adversary [52,41], our methods is more efficient and effective in principle, because (1) we do not require extra samples from the environment; (2) learning an RL adversary needs to explore the best perturbations to minimize the long-term reward of the victim, while we avoid such exploration by directly computing the worst-attack value with convex relaxation of victim policy; (3) our algorithm evaluates a lower bound of the worst-case policy value with convex relaxation, while learning an RL adversary usually results in an upper bound unless the globally optimal adversary is found.

---

### Official Review · Reviewer_z1xW · 2022-07-12

**Rating:** 7
**Confidence:** 3
**Soundness:** 3 good
**Presentation:** 3 good
**Contribution:** 3 good

**Summary:**

The paper introduces a robust training algorithm for RL. The key idea is to use a worst-case Bellman operator that allows calculation of the worst-case expected return of a policy under an adversarial attack. The new value function can then be used to find policies via both value-based methods or policy gradient ones. The authors include a weighting scheme to guide NN robustness towards more critical states.

**Questions:**

How necessary is $\kappa_{wst}$ for the convergence of the algorithm, for example in WorcA-DQN? Does $\kappa_{wst} = 0$ work?

How sensitive is the algorithm to the schedule of $\kappa_{wst}$?

**Limitations:**

There is not much discussion in this regard. More explanation would benefit the paper.

**Strengths And Weaknesses:**

The paper introduces a critical idea in robust training in RL. The proposed way of finding the robust policy by a value-based method directly and without a separate attacker module is significant. According to the paper's literature review, it has been a crucial idea missing in prior work. The authors do a good job empirically evaluating the new algorithm compared to prior work. The fact that the method can be applied to many existing algorithms is a nice property. Overall, the paper is well-written and impactful.

As a weakness, I wish the paper included some theoretical insight. Knowing the dynamics of the new operator even in finite MDPs is interesting. For example, the convergence of worst-case values for a fixed policy is intuitive and established, but when the policy is being updated, the dynamics are not as clear.

Related to uncertainty in non-stationary policy case, I wonder if it is necessary to combine robust values with natural ones via $\kappa_{wst}$. In WorcA-DQN, the algorithm uses worst-cases values for most of the training phase and then it is increased. A more discussion about such design choices would be appreciated.

==== Post Rebuttal ===
I thank the authors for their clarifications on my concerns. My comment on $\kappa_{wst}$ was properly answered. However, my score remains the same.

---

> ### Author Response · Authors · 2022-08-01
> **[2/2] About the impact and schedule of $\kappa_{wst}$**
>
> $\newcommand{zxw}{\textcolor{green}{\mathrm{z1xW}}}$
> > Q3: How necessary is $\kappa_{wst}$ for the convergence of the algorithm? How sensitive is the algorithm to the schedule of $\kappa_{wst}$?
>
> As we mentioned in the answer to Q2, the choice of the worst-case value's weight $\kappa_{wst}$ is **to control the trade-off between the final natural performance and robustness**. It does not have much influence on the convergence of the algorithm. We show the performance of our algorithm with various values of $\kappa_{wst}$ in Figure 10 in Appendix D.3.5. When we increase the weight of worst-case values $\kappa_{wst}$, the reward under worst-case perturbations increases, but it leads to a reduction of the natural reward. On the other hand, when $\kappa_{wst}$ is set close to 0, the algorithm is similar to standard training, where the policy achieves high reward under no attack, but extremely low reward under attacks. Hence, $\kappa_{wst}$ is necessary for our algorithm to control these two kinds of performance. In practice, one can adjust $\kappa_{wst}$ according to their preferences for robustness or natural performance.
>
> About the schedule of $\kappa_{wst}$, we provide our experimental setting in Appendix D.1.1 and D.1.2. In our experiment observation, WocaR-DQN is not very sensitive to the schedule, but WocaR-PPO needs a warm-up process with $\kappa_{wst}=0$ (the standard PPO without $L_{wst}$) to ensure efficient convergence. Without this warm-up, it may take more iterations to converge, but the final results are not affected much.
>
> > Q4: (Limitations) There is not much discussion in this regard. More explanation would benefit the paper.
>
> We appreciate the suggestion of the reviewer and have added the discussion of limitations in Appendix E.
>
> ---
>
> We again thank Reviewer $\zxw$ for reviewing our paper and giving constructive suggestions. We hope our above explanation has addressed all questions the reviewer has. We are happy to answer any further questions or concerns.
>
> Paper5837 Authors
>
> ---
>
> References:
>
>
> [1] Tsipras et al. Robustness may be at odds with accuracy. NeurIPS 2019.
>
> [2] Zhang et al. Theoretically principled trade-off between robustness and accuracy. ICML 2019.
>
> [3] Yang et al. A Closer Look at Accuracy vs. Robustness. NeurIPS 2020.
>
> [4] Mehrabi et al. Fundamental Tradeoffs in Distributionally Adversarial Training. ICML 2021.
>
> [41] Sun et al. Who is the strongest enemy? towards optimal and efficient evasion attacks in deep rl. ICLR 2022.
>
> [52] Zhang et al. Robust reinforcement learning on state observations with learned optimal adversary. ICLR 2021.

---

> ### Author Response · Authors · 2022-08-01
> **[1/2] About theoretical insights on convergence and the combination of robust values with natural ones**
>
> $\newcommand{zxw}{\textcolor{green}{\mathrm{z1xW}}}$
>
> We thank Reviewer $\zxw$'s valuable feedback and constructive suggestions. We are encouraged that Reviewer $\zxw$ finds our idea critical and our empirical results promising.
>
> For the questions raised by the reviewer, we have updated the manuscript to include more details, and we provide more explanations below.
>
> > Q1: As a weakness, I wish the paper included some theoretical insight. Knowing the dynamics of the new operator even in finite MDPs is interesting. For example, the convergence of worst-case values for a fixed policy is intuitive and established, but when the policy is being updated, the dynamics are not as clear.
>
> It is a good question. Our operator can evaluate the worst-case value of a fixed policy, but the policy gets updated during training. In experiments, we find that it is sufficiently good to have a rough approximation of the worst-case value at every step. When the policy gradually converges, the worst-case value could also gradually converge. Our insight is similar to the actor-critic framework where the critic is supposed to be $V^\pi$ but is not trained to convergence at every step. Instead, the actor and the critic are jointly trained. Another example is the ATLA[52] or PA-ATLA[41] method that alternately trains the victim and an "optimal" attacker. Although their definition of an optimal attacker is for a fixed victim, their implementation also gradually updates both the victim and the attacker, which works well in practice.
>
> This paper focuses on the robustness of deep RL models where a general convergence theory is not established yet, so we mainly evaluate the method in experiments. But we agree that a more rigorous analysis on the convergence of the proposed operator for a changing policy, even for finite MDPs and simple policy approximators, would be interesting and important. We hypothesize that it is related to the convergence theory of a two-player zero-sum game, and we will further investigate it in our future work.
>
>
> > Q2: Related to uncertainty in non-stationary policy case, I wonder if it is necessary to combine robust values with natural ones via $\kappa_{wst}$. In WorcA-DQN, the algorithm uses worst-cases values for most of the training phase and then it is increased. A more discussion about such design choices would be appreciated.
>
> We combine robust values with natural ones aiming to simultaneously improve the worst-case and natural performance. Our motivation for the design of two values comes from two perspectives.
> 1. From a geometric perspective, our Appendix B provides a geometric understanding of relations between the value perturbation polytope and policy robustness, which shows the changes of the worst-case values and the natural ones during training. We need to emphasize that enhancing the worst-case values doesn't necessarily improve the natural performance simultaneously. Besides effectively improving robustness, we hope that the agent can maintain comparable natural performance by also looking at the natural policy value.
> 2. Another perspective is the trade-off between robustness and accuracy that serves as a guiding principle in the design of supervised adversarial learning. For example, the well-known adversarial training method TRADES [2] designs a loss by combining the natural error and the robust error. This trade-off problem has been widely studied theoretically and empirically in existing robust learning works [1,2,3,4], but rarely has been discussed in RL setting. In our empirical experiments, we use $\kappa_{wst}$ to balance the natural and worst-case performance (robustness), as shown in Appendix D.3.5.
>
> We further explain the role of $\kappa_{wst}$ in detail in the answer to Q3 in the following comment.

---

### Official Review · Reviewer_9hAu · 2022-07-16

**Rating:** 6
**Confidence:** 4
**Soundness:** 2 fair
**Presentation:** 3 good
**Contribution:** 3 good

**Summary:**

This paper proposes a strong and efficient robust training framework for RL to directly estimate the optimizes the worst-case reward of a policy under p attacks. The authors empirically show that the proposed robust RL framework achieves the STOA results under strong attacks.

**Questions:**

It is great to see that empirically the proposed robust RL framework achieves high robustness. Is there any analytical results for the robustness? say, the lower bound of the robustness as a function of the convexity approximation of the worst-case attack？
At the beginning of the paper, the authors mentioned that the explanation results of the proposed framework make sense, while it is not evaluated in the evaluation section. Are there interesting findings and observations on general games regarding the explanation of the proposed framework?

**Limitations:**

The technical novelty of the paper is limited due to the direct estimation of the worst-case adversary/value. Maybe it is helpful for the authors to clarify the technical novelties of the paper, especially compared to [54,41] in addition to the conceptual differences.

**Strengths And Weaknesses:**

The paper addresses an important problem of robust RL, and the proposed method is clear and the paper is well-organized. Some analysis such as the robustness explanation of the proposed method is interesting.

However, the technical novelty of the paper is limited. For instance, the worst-case bellman operator is similar to the worse-case attack defined in [54, 41]. The worst case value estimation should be the most interesting part of the paper, while it’s just based on standard convex estimation for the worst case attack, but the DNN based RL the Q function is not convex and there are lots of strong assumptions in this process, which makes the estimation less valuable.

---

> ### Author Response · Authors · 2022-07-28
> **[2/2] More Explanation on Interpretation Results**
>
> $\newcommand{hau}{\textcolor{red}{\mathrm{9hAu}}}$
>
> > Q3: It is great to see that empirically the proposed robust RL framework achieves high robustness. Is there any analytical results for the robustness? say, the lower bound of the robustness as a function of the convexity approximation of the worst-case attack？
>
> Yes, our proposed framework evaluates and improves a lower bound of the robustness. Please see our answer to Q2 for more details. Our analysis in Appendix B and Appendix C explains this lower bound of robustness from multiple perspectives.
>
>
> > Q4: At the beginning of the paper, the authors mentioned that the explanation results of the proposed framework make sense, while it is not evaluated in the evaluation section. Are there interesting findings and observations on general games regarding the explanation of the proposed framework?
>
>
> If the reviewer is referring to the interpretation of agent behaviors (Figure 2), we have **more gif demonstrations** on MuJoCo environments in supplementary materials. As in prior work, our experiment section mainly focuses on the robustness and efficiency of the proposed method. We provide the behavior visualization in Figure 2 to point out a possible overfitting issue of attack-driven robust training methods. In the Walker environment, PA-ATLA-PPO agent learns to move forward by jumping with the right leg to avoid attacks on the left leg. Therefore, **attack-driven ATLA methods may overfit to some specific type of attack**, because they explicitly train an adversary and let the agent adapt to the adversary. On the contrary, our WocaR-RL is not trained against any specific adversary, and it finally learns to lower down its body, which is robust against a variety of attack algorithms, as shown in our experimental results.
>
> In our observation, the overfitting problem of PA-ATLA exists in Hopper and Walker among the MuJoCo tasks we evaluated.  Studying the actual behaviors of trained robust agents in general games is an interesting problem to investigate in further work, although it is out of the scope of this paper.
>
>
> ---
>
> We again thank Reviewer $\hau$ for the time and effort in reviewing our paper. We hope that our explanations above have addressed all concerns of Reviewer $\hau$. If so, we would greatly appreciate it if Reviewer $\hau$ could consider raising their score. We are happy to answer any further questions.
>
> Paper5837 Authors
>
> ---
>
> References:
>
> [13] Gowal et al. On the effectiveness of interval bound propagation for training verifiably robust models. NeurIPS 2018.
>
> [41] Sun et al. Who is the strongest enemy? towards optimal and efficient evasion attacks in deep rl. ICLR 2022.
>
> [52] Zhang et al. Robust reinforcement learning on state observations with learned optimal adversary. ICLR 2021.
>
> [54] Zhang et al. Robust deep reinforcement learning against adversarial perturbations on state observations. NeurIPS 2020.

---

> > ### Comment · Reviewer_9hAu · 2022-08-09
> > **post rebuttal**
> >
> > Thank the authors for the detailed responses and my questions are addressed. I will increase my score.

---

> ### Author Response · Authors · 2022-07-28
> **[1/2] Clarification on Technical Novelty and Convex Relaxation**
>
> $\newcommand{hau}{\textcolor{red}{\mathrm{9hAu}}}$
>
> We appreciate Reviewer $\hau$'s feedback and review. We are encouraged that Reviewer $\hau$ finds our proposed framework strong and the paper well-organized. Due to some misinterpretation of our paper, Reviewer $\hau$ has a concern about our novelty and the convex relaxation technique. We would like to clarify this misinterpretation and address Reviewer $\hau$'s questions below.
>
> > Q1: The technical novelty of the paper is limited. For instance, the worst-case bellman operator is similar to the worst-case attack defined in [54, 41].
>
>
> We emphasize that our worst-attack Bellman operator is **different from** the optimal-adversary Bellman operator in prior works [54,41]. The main differences are as follows.
> 1. Eq (20) in [54] defines a Bellman contraction for the optimal adversary. But please note that their contraction considers the worst-case state perturbation $s_ν \in B(s)$ in the possibly large state space. In contrast, our Eq (1) computes the worst-case action perturbation $a \in A_{adv}$, where $A_{adv}$ is the set of actions possibly misled by the state adversary. In general, the action space is much smaller than the state space. So our method is more **scalable** to large-scale environments such as Atari.
> 2. More importantly, the Bellman contractions in [54] and [41] are from the adversary's point of view, and solving them requires learning a state adversary, which doubles the number of environment samples required as well as computational cost. In contrast, our worst-attack Bellman operator computes the worst-case value **from the victim's own perspective**, and it can be **efficiently** solved by existing samples collected by the victim during its training, **without the need of learning an RL adversary**. Line 221-233 explains more advantages of our method compared to learning an adversary.
>
>
> Indeed, there exists a nice equivalent relation between our proposed Bellman operator and the prior formulations [54, 41] as proven in Appendix A. This is because their ultimate goals are the same: evaluating the worst-case value of a victim policy. However, **their formulations and their ways of solving these Bellman equations are very different**. Our proposed formulation views the problem from a different and novel perspective as mentioned above, and thus our Bellman operator is more scalable and efficient for solving. Because of this novel formulation, our method significantly outperforms prior SOTA [54,52,41] in both efficiency and effectiveness.
>
> **Other contributions:** In addition to evaluating the worst-case value by the proposed Bellman operator (Mechanism 1), we also propose a way of optimizing the policy based on worst-attack critic (Mechanism 2), as well as a more advanced regularizer considering various state importance (Mechanism 3). In addition, Appendix B provides a geometric interpretation for our framework and related work, which sheds light on a deeper understanding of robust RL methods. **We believe that these contributions are significant and novel.**
>
>
> > Q2: The worst case value estimation should be the most interesting part of the paper, while it’s just based on standard convex estimation for the worst case attack, but in the DNN-based RL, the Q function is not convex and there are lots of strong assumptions in this process, which makes the estimation less valuable.
>
> Reviewer $\hau$ might have misunderstood our method as convex approximation. But we respectfully point out that we solve the $A_{adv}$ set using **convex relaxation**, or interval bound propagation (IBP [13]), which uses convex set to **rigorously bound** all possible outputs resulting from input perturbations for neural networks. This convex relaxation is rigorous for non-convex neural networks, although may be pessimistic as certain points in the convex set may not be attainable by the input perturbations. (Please kindly refer to Figure 2 in [13] for a visualization of the convex relaxation.)
>
>
> Therefore, in our method, all actions the agent could possibly select under state attacks are **guaranteed** to be in $A_{adv}$ computed by convex relaxation. As a result, the fixed point of Eq (1) with convex relaxation is **guaranteed to be a lower bound** of the worst-case value of the agent.

---

> ### Author Response · Authors · 2022-08-06
> **Does our response address your questions?**
>
> $\newcommand{hau}{\textcolor{red}{\mathrm{9hAu}}}$
>
> Dear Reviewer $\hau$,
>
> Thank you again for your thoughtful review. Does our response address your questions? We would appreciate the opportunity to engage further if needed.

---

> ### Author Response · Authors · 2022-08-09
> **Would you mind reading our rebuttal and letting us know your thoughts?**
>
> $\newcommand{hau}{\textcolor{red}{\mathrm{9hAu}}}$
>
> Dear Reviewer $\hau$,
>
> Thank you again for reviewing our paper! We would like to politely remind you that we have addressed all your concerns and questions in our former responses. Since the discussion period is ending soon, we would greatly appreciate it if you could let us know your thoughts about our rebuttal and update the rating. We are very happy to address any further questions you may have.
>
> Best,
>
> Paper5837 Authors

---

### Author Response · Authors · 2022-08-01
**General response to all reviewers and a summary of revisions**

$\newcommand{hau}{\textcolor{red}{\mathrm{9hAu}}}$
$\newcommand{nfev}{\textcolor{blue}{\mathrm{NfEV}}}$
$\newcommand{zxw}{\textcolor{green}{\mathrm{z1xW}}}$
$\newcommand{jo}{\textcolor{orange}{\mathrm{J3o4}}}$

We thank all reviewers for their valuable feedback and insightful questions. We are particularly encouraged that they consider the proposed method novel and crucial for robust training in RL ($\zxw$, $\nfev$, $\jo$), the empirical evaluation good and extensive ($\zxw$, $\jo$), the performance achieving state-of-the-art ($\hau$, $\nfev$) and our paper well-written ($\hau$, $\zxw$, $\nfev$, $\jo$).

We address individual questions of reviewers in separate responses. We have also uploaded a modified version of our paper based on reviewers' suggestions. The major changes are highlighted as red in the paper. In particular, we added more explanations of the convex relaxation ($\hau$, $\jo$), the description of computational cost in continuous action spaces ($\nfev$), and more comparison to prior work ($\hau$, $\jo$) in Section 4. We also added Appendix E which discusses the limitations and the social impacts of this work in detail ($\zxw$, $\jo$).


We greatly appreciate all reviewers' time and effort in reviewing our paper. We hope that our paper updates and responses have addressed all reviewers' questions and concerns. Please let us know if there are further questions.

Paper5837 Authors

---

### Meta-Review · Area_Chair_zcN3 · 2022-08-24

**Recommendation:** Accept
**Confidence:** Certain

**Metareview:**

This paper introduces a novel adversarial training method that directly computes a worst-case performance under budget bounded attacks during the training process. As a result, the method is more sample efficient and achieves state-of-the-art performance across a number of test cases.

The reviewers agree that the contributions are novel and well validated, making this paper a clear acceptance.


**Award:**

No

---

### Decision · Program_Chairs · 2022-09-14

Accept